# Epidemiology and Risk Factors of Delayed Sputum Smear Conversion in Malaysian Aborigines with Smear-Positive Pulmonary Tuberculosis

**DOI:** 10.3390/ijerph19042365

**Published:** 2022-02-18

**Authors:** Muhammad Naim Ibrahim, Nik Rosmawati Nik Husain, Aziah Daud, Thilaka Chinnayah

**Affiliations:** 1Department of Community Medicine, School of Medical Sciences, Universiti Sains Malaysia, Kota Bharu 16150, Kelantan, Malaysia; naemibr7@gmail.com (M.N.I.); aziahkb@usm.my (A.D.); 2TB and Leprosy Control Sector, Disease Control Division, Ministry of Health Malaysia, Putrajaya 62590, Malaysia; jdrthilaka@moh.gov.my

**Keywords:** delayed sputum smear conversion, smear-positive pulmonary TB, Malaysian aborigines, factors associated

## Abstract

Background: Tuberculosis (TB) remains a serious public health challenge despite enormous eradication efforts. Indigenous groups worldwide have a higher TB incidence and associated delayed sputum–smear conversion. The aim of this case–control study was to determine the epidemiology and factors associated with delayed sputum–smear conversion among Malaysian aborigines. Methods: We used secondary data from 2016 to 2020 in the MyTB surveillance system. Malaysian aborigines with smear-positive pulmonary TB were enrolled and followed until the end of the intensive phase. Descriptive statistics and multiple logistic regression were used for data analysis. Results: Of 725 Malaysian aborigines with pulmonary TB, 572 (78.9%) were smear-positive and 487 (78.9%) fulfilled the study criteria. The mean (SD) age of smear-positive pulmonary TB was 39.20 (16.33) years. Majority of participants were male (63%), Senoi tribe (54.9%), living in rural areas (88.1%), formally educated (60.4%) and living below the poverty line (97.1%). Overall, 93 (19.1%) of 487 patients showed delayed sputum-smear conversion and significantly associated factors, such as smoking (AdjOR: 3.25; 95% CI: 1.88, 5.59), diabetes mellitus (AdjOR: 12.84; 95% CI: 6.33, 26.06), and HIV infection (AdjOR: 9.76; 95% CI: 3.01, 31.65). Conclusions: Stakeholders should adopt targeted approaches to tackle the problem of aboriginal groups with pulmonary TB and these associated risk factors to realise the End TB target.

## 1. Introduction

Tuberculosis (TB) has been a significant disease ever since ancient times. According to the World Health Organization (WHO), TB kills 1.5 million people annually, making TB one of the leading mortality causes globally [1]. Therefore, even the enormous challenges posed by the COVID-19 pandemic should not dampen efforts to fight TB. The TB incidence in Malaysia was successfully reduced from 350 per 100,000 in 1961 to 100 per 100,000 population in 1980, but this level has remained unchanged since then. The most recent figure of 92 per 100,000 population in 2019 positions Malaysia as an intermediate TB burden country [2,3]. 

TB is caused by *Mycobacterium tuberculosis*, and 85% of the cases are primarily pulmonary infections, while the remaining 15% are extrapulmonary [4]. Pulmonary TB is further classified by the sputum–smear status as either smear-positive or smear-negative pulmonary TB. The smear-positive type accounts for 55% of pulmonary TB cases [3]. It is a hazard that can infect others via droplets or airborne transmission, thereby representing a potential source of infection in the community [5].

A newly diagnosed patient with pulmonary TB is recommended to comply with a six-month treatment regimen with the Directly Observed Treatment, Short-course (DOTS). This involves daily drug administration for two months of the intensive phase and four months of the maintenance phase [6]. The patient is also required to undergo periodic follow-up to assess the treatment response. Specifically, for the monitoring of sputum conversion, a repeat sputum smear is done at the end of the intensive phase. Sputum–smear conversion is the indicator recommended by the WHO for monitoring patient progress and is used as the performance indicator for the Malaysian national TB control programme [7]. The target for the sputum conversion rate is above 85%, and Malaysia has achieved this in the past few years [8], although some patients, immunocompromised patients, continue to have persistent smear-positive results that require attention. Despite this, several contributing factors involving background and locality, such as geographical area and minorities like aboriginal groups, are still not well explored. 

Malaysian aborigines, known as Orang Asli, are an indigenous people living across the Malaysian peninsula; they have a population of 178,197 and account for 0.5% of Malaysia’s population [9]. They consist of three main tribes: the Senoi, the largest tribe, followed by the Melayu-Proto and the Negrito [10]. Today, they receive better attention from the authorities and have better access to healthcare, education, and other benefits. However, they are still regarded as vulnerable groups, as most of them still live below the poverty line, with 33% living in hardcore poverty [11]. These conditions have led to poor health outcomes, resulting in a higher incidence of infectious disease than is seen in the general population [12,13]. Several studies conducted among indigenous peoples worldwide, such as those in Australia, Papua, Southern China, India and Africa, have reported an unacceptably higher TB incidence, morbidity and mortality in indigenous persons compared to the dominant populations [14,15,16,17,18]. The Orang Asli share similar characteristics with indigenous people from other countries and are vulnerable to TB and poor treatment outcomes. However, few studies have been conducted on their sputum–smear conversion rates and associated factors. 

The 2030 Agenda adopted by the United Nations Development Programme [19] is aimed at ensuring that no one will be left behind, and it endeavours to reach those furthest behind first to achieve its Sustainable Developmental Goals (SDG). Therefore, the development of a foundation for future reference is essential for recommending a national TB programme to policymakers for effective intervention measures. The aim of the current study was to explore the epidemiology of TB among Malaysian aborigines and to determine the factors associated with delayed sputum–smear conversion among Malaysian aborigines with TB from 2016 to 2020. 

## 2. Materials and Methods

### 2.1. Study Setting and Subjects Recruitment 

This study was conducted from December 2020 until the end of May 2021 utilising secondary data from the MyTB surveillance system from the TB and Leprosy Control Sector, Ministry of Health Malaysia. The study involved two parts: In the first part, we included all TB cases among Malaysian aborigines registered in MyTB from 1 January 2016 to 31 December 2020 to explore the TB burden in Malaysian aborigines. For the second part, we conducted a case–control study to determine the factors associated with delayed sputum–smear conversion among Malaysian aborigines. We included Malaysian aborigines with smear-positive pulmonary TB registered in MyTB from 1 January 2016 to 31 December 2020, who also fulfilled the study inclusion and exclusion criteria. The inclusion criteria are the case with delayed sputum smear conversion at the end of the intensive phase and the control group who had converted sputum smear at the end of the intensive phase. Smear positive TB patients that repeat sputum-smear testing after two months of TB treatment were included, and subjects with incomplete required data were excluded. The subjects and their characteristics were identified at the time of pulmonary TB diagnosis was made. The flowchart for the case–control study is shown in Figure 1. 

### 2.2. Study Population and Sample Size Determination

The number of subjects required for the case–control study was calculated using power and sample size (PS) software based on two independent proportions with the power of study set at 80% and type I error set at 5%. The ratio of subjects with delayed sputum-smear conversion versus sputum-smear conversion at the end of the intensive phase was set at 1 to 4. The sample size was estimated by allowing an additional 10% possibility of data error, giving a final required sample size of 476. No sampling was done since the total number of smear-positive pulmonary TB cases registered in the MyTB database from 1 January 2016 to 31 December 2020 and fulfilled the study criteria was 487; thus, all samples were included

### 2.3. Operational Definitions

Malaysian aborigines are referred to as Orang Asli, a race subject to the Aboriginal Peoples Act 195 [10]. Smear-positive pulmonary TB is a case of pulmonary TB that fulfils one of the following: (i) two or more pretreatment sputum smears test positive for acid-fast bacillus (AFB) or (ii) one sputum smear is positive for AFB and chest radiographic abnormalities are determined by a physician as lung involvement, or (iii) one sputum smear is positive for AFB and at least one sputum culture is positive for *M. tuberculosis* [6]. 

Delayed sputum–smear conversion is defined as smear-positive pulmonary TB with a sputum sample that remains AFB smear-positive at the end of the intensive phase [20]. The intensive phase is defined as the first two months of the treatment regimen with first-line anti-TB drugs [2]. Pretreatment sputum–smear grading is categorised based on the number of AFB seen microscopically. It is graded into four categories: scanty, 1+, 2+ and 3+ [3]. In the present study, it is graded as low grade (scanty and 1+) and high grade (2+ and 3+).

A residential area is categorised according to the Department of Statistics Malaysia (DOSM) [21]. The classifications are urban, where the total population is 10,000 or more and the majority are involved in non-agricultural activities, and rural, where the total population is less than 10,000 or the total population is 10,000 or more but the majority are involved in agricultural activities. Household income is also categorised according to the DOSM [21]. Above the poverty line is described as a household income above RM 2208, while below the poverty line is an income of RM 2208 and below.

### 2.4. Proforma

A proforma was used to extract and record the information on required variables from the MyTB surveillance system. It consists of three main components: sociodemographics (including age, sex, tribes, location of resides, education level and household income), lifestyle (consisting of smoking status) and clinical characteristics (consisting of underlying diabetes mellitus, HIV status, BCG scarring, treatment category, pretreatment sputum smear, pretreatment chest X-ray and treatment adherence). 

### 2.5. MyTB Surveillance System from TB and Leprosy Control Sector, Ministry of Health Malaysia

MyTB is an online tuberculosis surveillance system under the direction of the TB and Leprosy Control Sector, Disease Control Division, Ministry of Health Malaysia. The MyTB database consists of the patient’s sociodemographics, pre-treatment conditions, treatment and follow-up details, complications and treatment outcome, with a total of 127 variables. MyTB is input at the district health office level across Malaysia and monitored at the state health department and ministry of health levels. 

### 2.6. Statistical Analysis

Data were analysed using the Statistical Program for Social Sciences (SPSS) version 26.0 software (IBM, Armonk, NY, USA). The data were explored for any input errors and were cleaned. Any missing value was detected, and a preliminary data description was done. Descriptive analysis was conducted and presented as frequency (n) and percentage (%). The analysis involved the proportion of pulmonary TB cases followed by the proportion of smear-positive pulmonary TB cases among Malaysian aborigines with TB. Further descriptions were made based on the sociodemographics, lifestyle and clinical characteristics of the Malaysian aborigines with smear-positive pulmonary TB in Malaysia from 2016 to 2020. 

A simple and multiple (binary) logistic regression was applied to determine the association between delayed sputum–smear conversion and all 14 selected independent variables. The outcome of the study was the sputum–smear status at the end of the intensive phase, either sputum-smear converted or delayed sputum–smear conversion. The model was presented with an adjusted odds ratio, with a *p*-value at less than 0.05 and 95% confidence interval (95% CI) considered statistically significant. 

### 2.7. Ethical Consideration

The study was approved by the Human Research Ethics Committee of Universiti Sains Malaysia (Code: USM/JEPeM/20110587) and the Malaysia Medical Research Ethics Committee (MREC) (Code: NMRR-20-2622-57268 (IIR)). The study also obtained permission from the TB and Leprosy Sector, Disease Control Division, Ministry of Health Malaysia (Code: KKM/600-56/3/2(32)JLO2. No patient consent was needed as this study only utilised secondary data and no conflicts of interest were declared.

## 3. Results

In total, 808 Malaysian aborigines were diagnosed with TB and registered in MyTB during the five study years (2016 to 2020), as shown in Table 1. Of those, 725 (89.7%) cases were pulmonary TB, while 83 (10.3%) cases were extrapulmonary TB. Of registered TB cases, the proportion of pulmonary TB ranged between 86.4% and 93.5%. Of the total pulmonary TB diagnosed and registered, 572 (78.9%) cases were smear-positive, while 153 (21.1%) were smear-negative. Of the 572, only 487 (85.1%) cases fulfilled the study criteria and were included in the study for further analysis. The remaining 85 cases were excluded from the study due to incomplete data.

### 3.1. Descriptive Analysis

#### 3.1.1. Sociodemographic, Lifestyle and Clinical Characteristics

A total of 487 cases with smear-positive pulmonary TB were included in the final analysis. Their ages ranged between 6 months and 82 years old. The majority were male, Senoi tribe members and living in rural areas. Most of them had received a formal education, with the majority up to the primary level, followed by secondary and tertiary. Most had household incomes below the poverty line. Lifestyle characteristics were assessed in terms of smoking status; the majority did not smoke. In terms of clinical characteristics, a small proportion of the subjects had comorbidities, such as diabetes mellitus and TB-HIV co-infection. Most had BCG scarring and were classified as new TB cases. Most of the pre-treatment sputum smears for AFB were categorised as high (59.1%). The majority of subjects presented with chest X-ray features indicating minimal lesions (64.5%), and most of them adhered to the DOTS treatment. The details are shown in Table 2.

#### 3.1.2. Delayed Sputum-Smear Conversion

Of the total cases, 93 (19.1%) showed delayed sputum–smear conversion, whereas the other 394 (80.9%) cases successfully converted. The mean (SD) age of cases with delayed sputum conversion was 43.73 (15.82) years, whereas the mean age of those whose smears converted at the end of the intensive phase was 38.13 (16.29) years. Delayed sputum smear-conversion was slightly higher in male (19.9%), highest in the Melayu Proto tribe (26.4%), those living in the urban area (22.4%), and smokers (25.4%). The majority of cases with delayed sputum–smear conversion had diabetes mellitus (59.6%), HIV co-infection (57.1%), advanced pretreatment chest X-ray findings (66.7%), and not adhering to treatment (93.2%) compared to the controls group. Full details on the sputum–smear status at the end of the intensive phase are shown in Table 2.

### 3.2. Factors Associated with Delayed Sputum–Smear Conversion

Simple logistic regression analysis showed that age and the Melayu Proto tribe were statistically significantly associated with delayed sputum–smear conversion. The lifestyle and clinical characteristics associated with delayed sputum–smear conversion were smoking, diabetes mellitus, HIV infection, pre-treatment sputum–smear AFB and treatment adherence.

The variables selected from simple logistic regression analysis were further analysed using multiple logistic regression to identify the associated independent factors. Treatment adherence was not selected for further analysis due to a very large crude OR and a wide 95% CI found during simple logistic regression, indicating a small number of subjects with this characteristic.

The significant factors identified from the multivariable analysis were smoking (AdjOR: 3.25; 95% CI: 1.88, 5.59; *p* < 0.001), diabetes mellitus (AdjOR: 12.84; 95% CI: 6.33, 26.06; *p* < 0.001) and HIV infection (AdjOR: 9.76; 95% CI: 3.01, 31.65; *p* < 0.001). When other variables were controlled for, the significant risk factors for delayed sputum–smear conversion among Malaysian aborigines with pulmonary TB were smoking (3.25 times more likely), underlying diabetes mellitus (12.84 times more likely) and HIV co-infection (9.76 times more likely). The model was fit, as shown by the fitness test results, and could accurately discriminate 72.1% of the cases. The details of simple and multiple logistic regression analyses are shown in Table 3.

## 4. Discussion

### 4.1. Tuberculosis Epidemiology in Malaysian Aborigines

TB elimination is still a long way away. It is a chronic infectious disease, so eradication efforts for TB are not simply treatment and need to extend beyond patient factors. Based on the latest figures for the Malaysian aboriginal population provided in 2018, the incidence of TB in aborigines can be deduced as 107 per 100,000 population. The figure was slightly higher than the national incidence in 2018, at 92 per 100,000 population [22]. The current study showed that pulmonary TB was the most common type of TB among Malaysian aborigines, ranging between 86% and 96%. This situation is similar in the general Malaysian population, where 75% to 85% of TB cases in 2014 until 2018 were pulmonary TB [8]. Similar findings have also been reported for high-burden countries, such as Indonesia, and low burden countries, like Canada, where pulmonary TB is the commonest type of TB [23,24]. This can be related to the nature of TB transmission, as the bacilli are transmitted mainly by air and later acquired through inhalation and deposited into the lungs [3].

The current study found a reduction in new TB cases of 84.3%, from 198 cases in 2019 to 31 cases in 2020. Studies in China and the Philippines also showed reductions in new TB cases of up to 39% and 78.7%, respectively, due to COVID-19 management interventions like public transport restrictions, movement control orders, cancellation of community activities and diversion of resources [25,26]. In Malaysia, the movement control order enforcement led to a 9% case reduction in 2020 compared to 2019 [26]. Resource reductions in indigenous community health services have been reported in the United States of America due to diversion of priorities to COVID-19, which adversely impacted accessibility of those communities to healthcare [27]. This situation can be related to the situation of the Malaysian aborigines, who face challenges in comprehending the new norms (e.g., scanning QR codes), restriction of public transport and wearing masks. All of these challenges can limit their access to health services.

### 4.2. Delayed Sputum–Smear Conversion among Malaysian Aborigines

Overall, 93 (19.1%) patients had delayed sputum–smear conversion, and 394 (80.9%) patients were sputum–smear converted at the end of the intensive phase. The current study finding showed that the rate of delayed sputum–smear conversion is higher among Malaysian aborigines than at the national level (which has remained below 15.0% since 2010) [28]. A study among rural residents in Tirupati, India, reported that 46.0% had delayed sputum–smear conversions compared to the urban residents in that state. This was thought to be associated with poor treatment adherence due to poor health care accessibility [29]. Another study involving rural tribes in Porto Alegre, Brazil, reported that 35.0% had delayed sputum–smear conversion and this was thought to be associated with late presentation and a high proportion of drug-resistant TB [30]. However, studies on delayed sputum–smear conversion among Malaysian aborigines, and in the Southeast Asia region in general, are limited.

### 4.3. Risk Factors for Delayed Sputum–Smear Conversion among Malaysian Aborigines

The analysis showed that delayed sputum–smear conversion among Malaysian aborigines with smear-positive pulmonary TB was significantly associated with smoking, diabetes mellitus and HIV co-infection. The odds of smoking among individuals with delayed sputum–smear conversion was about 3.25 times that among the controls group when other confounders were adjusted. These findings are comparable with the research conducted in Brazil, Hong Kong and Sri Lanka [31,32,33]. In addition, a case–control study conducted in the rural area of Lampung, Indonesia, showed that smoker patients were at much higher risk (7.46 times more likely) for delayed sputum–smear conversion compared to non-smoker patients [34]. The hazardous substances in tobacco smoke are speculated to contribute to the delayed sputum–smear conversion in many ways. According to Aghapour et al. (2018) [35], tar, acrolein and formaldehyde are among identified irritants in smoking that cause inflammation and injury to the respiratory epithelium and submucosal secretory glands, leading to mucus hypersecretions and accumulation, as well as impairment of ciliary oscillations. These pathological changes lead to a delay in mucocilliary clearance, creating conditions favourable for colonisation by *Mycobacterium tuberculosis*. In addition, the oxidative stress chemical in smoking such as hydrogen peroxide, affects the efficacy of anti-TB medication by reducing the concentration and phosphorylation of antimicrobials, thereby contributing to a delay in sputum–smear conversion [36].

The current findings showed that Malaysian aborigines who had underlying diabetes mellitus with delayed sputum–smear conversion had the odd of 12.84 times than those controls group when other confounders were adjusted. These results are in line with much past research. For example, a case–control study conducted at the Institute of Respiratory Medicine, Kuala Lumpur, Malaysia, showed that TB patients with underlying diabetes mellitus were four times more likely to have delayed sputum–smear conversion [28]. Another study conducted in Taipei, Taiwan, reported that patients with underlying diabetes mellitus were 10.91 times more likely to have delayed sputum–smear conversion [37]. Being a diabetic patient leads to cumulative effects of chronic hyperglycaemia, and later reduces the expression signals from macrophages and dendritic cells and results in a poor innate immune response [38]. Patients with poorly controlled diabetes were reported to show a delay in the absorption of the anti-TB medication. Poor drug absorption was due to an impairment of the glycoprotein transporter in the intestine lining and a reduction in intestinal motility, leading to a poor immune response to eliminate the bacilli [39]. Thus, those patients were more likely to have a delayed sputum-smear conversion.

Current research revealed that the odds of HIV co-infection among individuals with delayed sputum-smear conversion is ~9.76 times that among the controls group when other confounders were adjusted. This finding was in agreement with a study in Mumbai, India, which showed that HIV co-infected TB patients took longer (a median of 20 weeks) than HIV-negative patients (a median time of 5 weeks) for sputum–smear conversion [40]. Another retrospective cohort study involving pulmonary TB patients in Kigali, Rwanda, found that HIV co-infected patients were three times more likely to show delayed sputum-smear conversion [41]. This was due to poor antiretroviral therapy enrolment, and most of these patients presented with late presentations. Clinical evidence has shown that TB patients with HIV co-infection are likely to be misdiagnosed, as commonly occurs among untreated HIV patients with high viral load and CD4+ counts below 200 cells per mm^3^ [42]. In addition, most of these patients present with non-specific symptoms, frequently having negative AFB sputum smears and atypical radiological presentations, thereby contributing to the late presentation of the disease [43]. However, the wide 95% confidence interval for the adjusted odds ratio shown in the current research did affect the study precision due to the small number of HIV co-infected patients.

The current study found a slightly older age for delayed sputum-smear conversion, with a mean (SD) age of 45.73 (15.82) years compared to the mean age of 38.13 (16.29) years for the patients with converted smears at the end of the intensive phase. However, age was not a statistically significant risk factor, in agreement with several studies from South Africa, Iran and Japan that reported delayed sputum–smear conversion was common in the older age group, but the multivariable analysis showed no significance [44,45,46]. Household income also showed no significant association with delayed sputum-smear conversion, in agreement with studies conducted in Lampung, Indonesia and Bangladesh, where household income among the subjects was not associated with delayed sputum–smear conversion and treatment outcome [34,47]. Many countries give low-income groups or populations incentives and special care, including TB treatment subsidies, healthcare provider outreach activities and collaboration with non-government organisations. Pretreatment sputum-smears and chest X-ray severities were also not statistically significantly associated with delayed sputum–smear conversion at the end of the intensive phase. These factors would reflect the bacterial load before the treatment and would supposedly require a longer period for bacterial clearance with medication [40]. Several studies have reported similar findings to those in the current study, where the factors were not significant for the timing of sputum conversion after the initiation of treatment [48,49].

### 4.4. Study Limitations

This study had several limitations. One was its use of secondary data and reliance on the availability of the MyTB surveillance database. Another was that several characteristics regarded as important and that could affect the study outcome were not available; these included factors such as alcohol consumption, substance abuse, body mass index and other comorbidities like anaemia, diabetes control level and chronic obstructive pulmonary disease. Besides that, the antiretroviral therapy (ART) status among HIV patients were not included in the study; thus, the logistic regression model analysis was not adjusted for ART. The data quality, such as self-reporting smoking status, was also subject to selection of response and recall bias. The low prevalence of HIV co-infection among Orang Asli also led to a wide confidence interval during the analysis. Finally, the reduction in case detections due to the COVID-19 pandemic resulted in a small sample in the later years.

### 4.5. Future Research

Future research should be conducted using primary data collection, as this provides more options for variables and better data quality. Quantitative and qualitative methods can explore other factors, such as knowledge, attitude and practices in delayed sputum-smear conversion among aboriginal patients. Genetic studies would also be interesting for exploration of these associations. These studies could improve our understanding of the disease among aborigines, thereby improving the current TB programme for achieving the End TB target.

## 5. Conclusions

Delayed sputum–smear conversion among Malaysian aborigines was significantly associated with smoking, diabetes mellitus and HIV co-infection. Smokers with smear-positive pulmonary TB should be managed thoroughly, and stop-smoking clinic services should be made accessible. A smoke-free role model and multi-agency collaboration are needed for promoting a smoke-free environment in the aboriginal settlements. Diabetes mellitus and HIV co-infection in smear-positive pulmonary TB among aboriginal populations need to be managed cautiously via interdisciplinary approaches. The circumstances for assessing health services must be actively managed by providing logistic and financial support. Malaysian aborigines have a unique culture and traditions, so one-size-fits-all programmes are not the solution. The national TB programme should be tailored according to their background.

## Figures and Tables

**Figure 1 ijerph-19-02365-f001:**
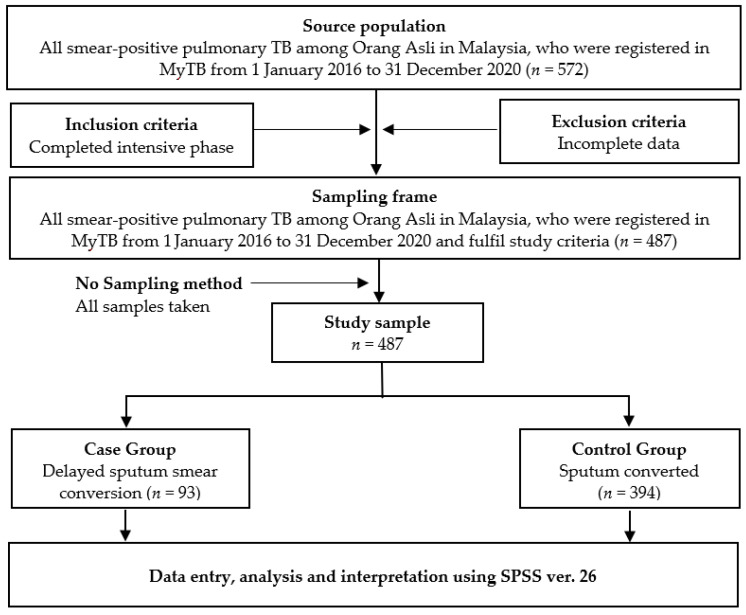
The flowchart for the case–control study.

**Table 1 ijerph-19-02365-t001:** Description of TB cases among Malaysian Aborigines from 2016 to 2020 (*n* = 808).

Year	TB Cases (*n* = 808)	Pulmonary TB Cases (*n* = 725)	Smear Positive Pulmonary TB (*n* = 572)	Proportion ^a^ (95% CI) ^b^
2016	177	161	118	0.73 (0.66–0.80)
2017	211	193	157	0.81 (0.75–0.87)
2018	191	165	134	0.81 (0.74–0.87)
2019	198	177	138	0.78 (0.71–0.84)
2020	31	29	25	0.86 (0.68–0.96)
Summative	808	725	572	0.79 (0.76–0.82)

**^a^** Proportion of smear-positive pulmonary TB/pulmonary TB; **^b^** 95% CI = 95% confidence interval.

**Table 2 ijerph-19-02365-t002:** The characteristics of smear-positive pulmonary TB and its sputum smear status at the end of the intensive phase among Malaysian aborigines from 2016 to 2020 (*n* = 487).

Variables	Frequency (%)	Delayed Sputum Smear Conversion (*n* = 93) *n* (%)	Sputum Smear Converted (*n* = 394) *n* (%)
**Sociodemographic**			
**Age (years)**	39.20 (16.33) *	43.73 (15.82) *	38.13 (16.29) *
**Sex**			
Female	180 (37.0)	32 (17.8)	148 (82.2)
Male	307 (63.0)	61 (19.9)	246 (80.1)
**Tribes**			
Senoi	300 (61.6)	50 (16.7)	250 (83.3)
Melayu Proto	121 (24.8)	32 (26.4)	89 (73.6)
Negrito	66 (13.6)	11 (16.7)	55 (83.3)
**Location of residence**			
Urban	58 (11.9)	13 (22.4)	45 (77.6)
Rural	429 (88.1)	80 (18.6)	349 (81.4)
**Education level**			
Tertiary	5 (1.0)	1 (20.0)	4 (80.0)
Secondary	110 (22.6)	18 (16.4)	92 (83.6)
Primary	179 (36.8)	32 (17.9)	147 (82.1)
Informal	193 (39.6)	42 (21.8)	151 (78.2)
**Household income**			
Above poverty line	14 (2.9)	3 (21.4)	11 (78.6)
Below poverty line	473 (97.1)	90 (19.0)	383 (81.0)
**Lifestyle characteristic**			
**Smoking**			
Non-smoker	259 (53.2)	35 (13.5)	224 (86.5)
Smoker	228 (46.8)	58 (25.4)	170 (74.6)
**Clinical characteristic**			
**Diabetes mellitus**			
No	440 (90.3)	65 (14.8)	375 (85.2)
Yes	47 (9.7)	28 (59.6)	19 (40.4)
**HIV status**			
No	473 (97.1)	85 (18.0)	388 (82.0)
Yes	14 (2.9)	8 (57.1)	6 (42.9)
**BCG scar**			
Present	371 (76.2)	76 (20.5)	295 (79.5)
No	116 (23.8)	17 (14.7)	99 (85.3)
**Treatment category**			
New case	462 (94.9)	89 (19.3)	373 (80.7)
Relapse	25 (5.1)	4 (16.0)	21 (84.0)
**Pretreatment sputum smear**			
Low	199 (40.9)	26 (13.1)	173 (86.9)
High	288 (59.1)	67 (23.3)	221 (76.7)
**Pretreatment chest X-ray**			
No lesion	27 (5.5)	0 (0.0)	27 (100.0)
Minimal	314 (64.6)	28 (8.9)	286 (91.1)
Moderate	137 (28.1)	59 (43.1)	78 (56.9)
Advance	9 (1.8)	6 (66.7)	3 (33.3)
**Treatment adherence**			
Yes	443 (91.0)	52 (11.7)	391 (88.3)
No	44 (9.0)	41 (93.2)	3 (6.8)

* Mean (Standard deviation).

**Table 3 ijerph-19-02365-t003:** Simple and multiple logistic regression analysis for factors associated with delayed sputum smear conversion.

Variables	Crude OR (95% CI)	*p*-Value ^a^	Adj. OR (95% CI)	*p*-Value ^b^
**Sociodemographic**				
**Age (years)**	1.02 (1.01, 1.04)	0.003		
**Sex**				
Female	1			
Male	1.15 (0.71, 1.84)	0.571		
**Tribes**				
Senoi	1			
Melayu Proto	1.78 (1.09, 2.98)	0.023		
Negrito	1.01 (0.49, 2.05)	0.998		
**Location of residence**				
Urban	1			
Rural	0.79 (0.41, 1.54)	0.494		
**Education level**				
Tertiary	1			
Secondary	0.78 (0.08, 7.42)	0.831		
Primary	0.87 (0.09, 8.05)	0.903		
Informal	1.11 (0.12, 10.22)	0.925		
**Household income**				
Above poverty line	1			
Below poverty line	0.86 (0.24, 3.15)	0.822		
**Lifestyle**				
**Smoking status**				
Non-smoker	1		1	
Smoker	2.18 (1.37, 3.47)	0.001	3.25 (1.88, 5.59)	<0.001
**Clinical**				
**Diabetes mellitus**				
No	1		1	
Yes	8.50 (4.49, 16.11)	<0.001	12.84 (6.33, 26.06)	<0.001
**HIV status**				
No	1		1	
Yes	6.09 (2.06, 18.00)	<0.001	9.76 (3.01, 31.65)	<0.001
**BCG scar**				
Present	1			
No	0.67 (0.38, 1.48)	0.365		
**Treatment category**				
New case	1			
Relapse	0.80 (0.27, 2.38)	0.686		
**Pretreatment sputum smear**				
Low	1	0.005		
High	2.02 (1.23, 3.31)			
**Pretreatment chest X-ray**				
No lesion	1	0.525		
Minimal	1.05 (0.83, 2.92)	0.644		
Moderate	1.49 (0.94, 3.45)	0.505		
Advance	1.37 (0.84, 2.05)			
**Treatment adherence**				
Yes	1	<0.001		
No	102.76 (30.72, 343.73)			

^a^ Simple logistic regression; ^b^ Multiple logistic regression; 95% CI = 95% confidence interval; Forward LR and Backward LR method applied; No multicollinearity and no interaction were found; Hosmer and Lemeshow test, *p*-value = 0.212; The classification table is 83.4% correctly classified; The area under the receiver operating characteristic (ROC) curve was 72.1%.

## Data Availability

The data presented in this study are available on request from the corresponding author. The data are not publicly available due to privacy and confidentiality.

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
