# Peer review of "Epidemiology and Risk Factors of Delayed Sputum Smear Conversion in Malaysian Aborigines with Smear-Positive Pulmonary Tuberculosis"

_ijerph, 2022, doi:10.3390/ijerph19042365_

Round 1

Reviewer 1 Report

General comments

Remove all the subheadings in the abstract.

Specific comments

Line 25: with pulmonary….? Something is lacking.

Line 33: death

Line 37: … 1961 to 100 per …….in  1980. Something is lacking.

Lines 54-58: Rephrase to “The target for the sputum conversion rate is above 85%, and 54 Malaysia has achieved this in the past few years [8], although some patients, immunocompromised patients, continue to have persistent smear-positive results that require attention. Despite this, several contributing factors involving background and locality, such as geographical area and minorities like aboriginal groups, are still not well explored.”

Line 65: hardcore

Line 198: Please situate this figure within the text before placing it here. You cannot just pop up a figure without indicating what it means.

Line 294: 198?

Line 320-336: It would be interesting to mention some of the specific cigarette components and how they affect the smokers’ health and contribute to increased TB. Just mentioning toxic substances may not be sufficient,

Author Response

Thank you for the kind and generous comments. Below is the response to the comments.

Point 1: Line 25: with pulmonary….? Something is lacking.

Response 1: There is a sentence duplication with the conclusion statement below it. Correction is made by deleting the mentioned hanging statement (Line 24 to 25).

Point 2: Line 33: death

Response 2: Correction made from deaths to ‘death’ (Line 32).

Point 3: Line 37: … 1961 to 100 per …….in  1980. Something is lacking.

Response 3: Correction was made as below. (Line 36 & 37).

 ….1961 to 100 per 100,000 population in 1980, but this level has remained unchanged since then. The most recent figure of 92 per 100,000 population in 2019 positions …

Point 4: Lines 54-58: Rephrase to “The target for the sputum conversion rate is above 85%, and 54 Malaysia has achieved this in the past few years [8], although some patients, immunocompromised patients, continue to have persistent smear-positive results that require attention. Despite this, several contributing factors involving background and locality, such as geographical area and minorities like aboriginal groups, are still not well explored.”

Response 4: The statements were rephrased as suggested. (Line 53 – 58). 

Point 5: hardcore

Response 5: The spelling was corrected from ‘hard-core’ to ‘hardcore’ (Line 65).

Point 6: Line 198: Please situate this figure within the text before placing it here. You cannot just pop up a figure without indicating what it means.

Response 6: Figure 1 is mentioned within the text under subsection 2.1: Study Setting and Subject Recruitment. (Line 93-94).

Point 7: Line 294: 198?

Response 7: It is a correct figure, indicating 198 cases. Thus, to avoid confusion, the sentence was improved as below. (Line 290-291).

The current study found a reduction in new TB cases of 84.3%, from 198 cases in 2019 to 31 cases in 2020.

Point 8: Line 320-336: It would be interesting to mention some of the specific cigarette components and how they affect the smokers’ health and contribute to increased TB. Just mentioning toxic substances may not be sufficient,

Response 8: Two sentences were improved as suggested below. Examples of cigarette components that cause disruption in normal physiology like irritants (tar, acrolein and formaldehyde)  and oxidative stress (hydrogen peroxide) were added.

Line 326 – 327: According to Aghapour et al. (2018), tar, acrolein, and formaldehyde are amongst identified irritants in smoking that cause …..

Line 331 – 332: In addition, the oxidative stress chemical in smoking such as hydrogen peroxide, affects the efficacy of …..

Reviewer 2 Report

Aboriginal TB smear conversion

This manuscript reports smear-conversion among Malaysian aborigines

with smear positive TB between 2016 and 2020 who completed intensive phase of treatment. The study was interesting that it targeted the important segment of the population. However, the main drawback of the work is that it the research question, study design and analysis plan are not in synchrony. Although the authors reported of conducting a retrospective study, the sample size calculation and the reported results as well as the reporting fits into a case-control design (Example – Table 2 and use of logistic regression). This confusion makes reading the paper challenging and needs fixing. The second major issue is the selective reporting of selected results despite the fact that these variables are similar between smear converted and smear-unconverted population. This is misleading as it carries implicit concept of causation.  The study would have been much interesting if the authors have included data from other ethnic groups and showed if ethnic groups(tribes) and all other variables predicted the outcome.  

Therefore, the manuscript would benefit from clear study design which is synchronised with appropriate analysis plans and reporting. In its current form its hard to provide feedback as these are not clear in the manuscript. Please see below for detailed feedback.

Overall presentation

The writing style has room for improvement. In several sections including the introduction, results and discussion the manuscript reports results that are not among the stated study objectives. Example – Lines 280-293, Lines 201-208,

Sample size

The stated design is retrospective cohort although the sample size calculation used looks more of a case-control study. This is just completely difficult to understand. The study recruited just one group of population and sample size is calculated for two population proportion. Although the assumed ratio is 3:1, the final sample size is 4.24:1. This should be clarified. Please provide the formula used and all other inputs to it.

Before calculating sample size, the authors need to make sure what they want to achieve. If the aim is to see the incidence and relative risk of the outcome, cohort study design would be ok and sample size should done accordingly. If the interest is to see the predictors of delayed smear conversion, case-control study would have been ok. However, if the interest is both on the magnitude of delayed conversion and risk factors then cross-sectional (retrospective cross-sectional) would be ok. This distinction important because it is the crucial step to decide the statistical analysis and the result reporting. In the current format- things are mixed up- the design is retrospective cohort and analysis is using logistic regression. This is not the right approach. Furthermore, the authors are reporting odds ratio as if they are risk ratios.  This has to be fixed.

Statistical analysis

Really hard to understand- example- the authors stated that (Lines 149-150)- The analysis involved the proportion of pulmonary TB cases among Malaysian 149 aborigines with TB. As the study started off with smear positive TB, this analysis plan does not make sense.

The authors also applied logistic regression to come up with predictors of delayed smear conversion. On lines 156-157- they stated adjusted relative risk was calculated from this model. Relative risk is not the direct output from Logistic regression. Logistic regression provides odds ratio.

Results

It’s hard to give appropriate feedback due to lack of clarity in the methods. However, some points are below

Selective reporting

The authors selective reporting can mislead the readers. For example- the authors stated in lines 231-233-

“The majority of smear- 231 positive pulmonary TB cases who ended up with delayed sputum-smear conversion were 232 men (63.0%), from the Senoi tribe (61.6%), and lived in rural areas (88.1%)”. This report sounds like delayed conversion is related to male sex, Senoi tribe and rural residence, although these figures are almost the same among those with successful smear conversion.  In contrast, individuals with successful conversion have higher proportion of Senoi tribe than those with delayed conversion rates. Moreover, the delayed category has higher Melayu Proto tribe (higher by more than 10 percentage points) compared to those in the successful conversion group.

Caution should be made in what is being communicated not to send wrong message.

Mismatch in reported numbers

 Line 21- Overall, 93 (19.1%) of 21 487 patients showed delayed sputum-smear conversion

Line 307- Overall, 84 (19.9%) patients had delayed sputum-smear conversion

Difficult conclusion given the study design

Lines 308-310- The current study finding showed that the rate of delayed sputum-smear conversion is higher among Malaysian aborigines than at the national level (which has remained below 15.0% ). This conclusion is hard given that the study design is not clear to us.

Author Response

The authors appreciate the kind and generous comments. Below is the response to the comments.

Point 1: In several sections including the introduction, results and discussion the manuscript reports results that are not among the stated study objectives. Example – Lines 280-293, Lines 201-208,

Response 1: The objective of the study was improved as below. (Line 78 -81).

The aim of the current study was to explore the epidemiology of TB among Malaysian aborigines and to determine the factors associated with delayed sputum-smear conversion among Malaysian aborigines with TB from 2016 to 2020.

Point 2: The study design - the authors need to make sure what they want to achieve. If the aim is to see the incidence and relative risk of the outcome, cohort study design would be ok and sample size should done accordingly. If the interest is to see the predictors of delayed smear conversion, case-control study would have been ok. However, if the interest is both on the magnitude of delayed conversion and risk factors then cross-sectional (retrospective cross-sectional) would be ok. This distinction important because it is the crucial step to decide the statistical analysis and the result reporting. In the current format- things are mixed up- the design is retrospective cohort and analysis is using logistic regression. This is not the right approach

Response 2: The study design was amended with a case-control design, instead of a retrospective cohort. We acknowledged the reviewer comments that the methods for sample size calculation and the statistical analysis using logistic regression is not aligned with the retrospective cohort study.

Hence, several changes have been made:

  • The term ‘retrospective cohort’ was replaced with a ‘case-control design’ (in the Abstract – Line 13, and in Section 2: Material and Methods - Line 84).
  • The term ‘cohort’ was replaced with ‘Majority of participants were…’ (In the Abstract - Line 20).
  • The sample size calculation is amended to suit the case-control study. The details are explained in Point 3 below.
  • The study flowchart is amended according to the case-control study (Figure 1 – Page 4).
  • The term ‘relative risk’ was corrected to ‘odds ratio’ (Line 154).

Point 3: Sample Size - The stated design is retrospective cohort although the sample size calculation used looks more of a case-control study. This is just completely difficult to understand. The study recruited just one group of population and sample size is calculated for two population proportion. Although the assumed ratio is 3:1, the final sample size is 4.24:1. This should be clarified. Please provide the formula used and all other inputs to it.

Response 3: The sample size calculation was corrected according to the case-control study design and following the comments by the reviewer. (Line 97 – 102).

The number of subjects required was calculated using power and sample size (PS) software based on two independent proportions with the power of study set at 80% and type I error set at 5%. The ratio of subjects with delayed sputum-smear conversion versus sputum-smear conversion at the end of the intensive phase was set at 1 to 4. The sample size was estimated by allowing an additional 10% possibility of data error, giving a final required sample size of 476.

Point 4: Statistical analysis - Really hard to understand- example- the authors stated that (Lines 149-150)- The analysis involved the proportion of pulmonary TB cases among Malaysian 149 aborigines with TB. As the study started off with smear positive TB, this analysis plan does not make sense.

Response 4: Correction was made accordingly for a better understanding of the flow of analysis conducted as below. (Line 145 - 146).

The analysis involved the proportion of pulmonary TB cases followed by the proportion of smear-positive pulmonary TB cases among Malaysian aborigines with TB.

Following that, the caption for Table 1 [Description of TB cases among Malaysian Aborigines from 2016 to 2020 (N=808)] also was improved to make it align with its content. (Line 203)

Point 5: Statistical analysis - The authors also applied logistic regression to come up with predictors of delayed smear conversion. On lines 156-157- they stated adjusted relative risk was calculated from this model. Relative risk is not the direct output from Logistic regression. Logistic regression provides odds ratio.

Response 5: Correction was made by replacing ‘adjusted relative risk’ with ‘adjusted odds ratio’ (Line 154).

Point 6: Selective reporting - The authors selective reporting can mislead the readers. For example- the authors stated in lines 231-233-

“The majority of smear- 231 positive pulmonary TB cases who ended up with delayed sputum-smear conversion were 232 men (63.0%), from the Senoi tribe (61.6%), and lived in rural areas (88.1%)”. This report sounds like delayed conversion is related to male sex, Senoi tribe and rural residence, although these figures are almost the same among those with successful smear conversion.  In contrast, individuals with successful conversion have higher proportion of Senoi tribe than those with delayed conversion rates. Moreover, the delayed category has higher Melayu Proto tribe (higher by more than 10 percentage points) compared to those in the successful conversion group.

Caution should be made in what is being communicated not to send wrong message.

Response 6: Authors acknowledge the miss interpretation of Table 2. The corrections were made as below. (Line 224 - 229).

Delayed sputum conversion was slightly higher in Male (19.9%), highest in the Melayu Proto tribe (26.4%), those living in the urban area (22.4%), and smokers (25.4%). The majority of patients with diabetes mellitus (59.6%), HIV co-infection (57.1%), advanced pre-treatment chest x-ray findings (66.7%), and not adhering to treatment (93.2%) had delayed sputum-smear conversion.  

Point 7: Mismatch in reported numbers

Line 21- Overall, 93 (19.1%) of 21 487 patients showed delayed sputum-smear conversion

Line 307- Overall, 84 (19.9%) patients had delayed sputum-smear conversion

Response 7: Authors acknowledge the mistake in writing the figure. The correction was made to ‘93 (19.1%)’. (Line 303).

Point 8: Difficult conclusion given the study design - Lines 308-310- The current study finding showed that the rate of delayed sputum-smear conversion is higher among Malaysian aborigines than at the national level (which has remained below 15.0%). This conclusion is hard given that the study design is not clear to us.

Response 8: The study design was corrected as explained in the response to Point 2.

Reviewer 3 Report

The percentages in table 2 are not correct, therefore the results are not correct either. The analysis must be performed again based on the correctly calculated results.

In Table 2, the percentages that were calculated in columns Delayed sputum smear conversion and Sputum smear converted are not correct. The percentage has to be calculated with the total of the first column (frequency). Here is an example made with the variable Pre-treatment sputum smear.

Calculation made by the authors

Table 2. The characteristics of smear-positive pulmonary TB and its sputum smear status at the end of the intensive phase among  Malaysian aborigines from 2016 to 2020 (n=487)

Variables

Frequency (%)

Delayed sputum smear conversion (n=93)

n (%)

Sputum smear converted (n=394)

n (%)

Pre-treatment sputum smear

Low

199 (40.9)

26 (28.0)

173 (43.9)

High

288 (59.1)

67 (72.0)

221 (56.1)

Correct calculation

Variables

Frequency (%)

Delayed sputum smear conversion (n=93)

n (%)

Sputum smear converted (n=394)

n (%)

Pre-treatment sputum smear

Low

199 (40.9)

26 (13.0)

173 (86.9)

High

288 (59.1)

67 (23.2)

221 (76.7)

The percentages in the first column of Table 2 are correct, but those in the other columns are not correct and have to be corrected

Author Response

The authors appreciate the kind and generous comments. Below is the response to the comments.

Point 1: The percentages in table 2 are not correct, therefore the results are not correct either. The analysis must be performed again based on the correctly calculated results.

In Table 2, the percentages that were calculated in columns Delayed sputum smear conversion and Sputum smear converted are not correct. The percentage has to be calculated with the total of the first column (frequency).

Response 1: The percentages in Table 2, as for column 2 (delayed sputum smear conversion) and column 3 (sputum smear converted) were corrected as suggested. With this correction,  we acknowledge that the figures are more meaningful to be explained and discuss in the later sections (result and discussion).

The corrected Table 2 is as below. (Page 6)

Round 2

Reviewer 2 Report

Thanks for taking time to respond to my suggestions.  The study has improved but there are issues that need fixing. Some of the incorporated responses were wrong direction

Major comments

  1. If the study design is a case-control study of factors associated with delayed smear conversion, then you can not answer other objectives. In a case-control study you just answer one question. In your case, using a case-control study design you can only answer one objective- that is factors associated with delayed sputum smear conversion. However, your aim statement on lines 86-88 "The aim of the current study was to explore the epidemiology of TB among Malaysian aborigines..." can't be answered by a case-control design. Again case-control study would start from smear conversion staus and look out for factors. In your study diagram (figure 1) all aborigines diagnosed with TB between 2016-2020 were included and there was no sampling involved. In this case you can remove your sampling section including sample size calculation and just provide the description of that (that all were included, no sampling applied etc). I, therefore, suggest removing the case-control study design from throughout the paper and just provide the description of the data and how that is captured into your study. 

2- I think this section (lines 253-255) needs rewriting - "The majority of patients with diabetes mellitus (59.6%), HIV co-infection (57.1%), advanced pre-treatment chest x-ray findings (66.7%), and not adhering to treatment (93.2%) had delayed sputum- smear conversion". It is incorrect because this study didn't study diabetic patients, HIV patients but it rather quantified diabetes and HIV among TB patients. So better to present the comparison of the proportion with diabetes or HIV among delayed converters and the controls. 

3. Is the HIV status in the logistic regression model adjusted for ART? if not please include in limitation of the study

4. Interpretation of results needs to based on the analysis. For example- On lines 471-473 the authors interpreted the odds ratio as - Malaysian aborigines with smear-positive pulmonary TB who smoked were 3.25 times more likely to have delayed sputum-smear conversion than non-smokers when other confounders were controlled. That is not quite right. This should be interpreted something like - the odds of smoking among individuals with delayed smear conversion is about 3.2 times that among controls. Please correct others to be consistent with your analysis(Example line 507)

Author Response

Point 1: If the study design is a case-control study of factors associated with delayed smear conversion, then you can not answer other objectives. In a case-control study you just answer one question. In your case, using a case-control study design you can only answer one objective- that is factors associated with delayed sputum smear conversion. However, your aim statement on lines 86-88 "The aim of the current study was to explore the epidemiology of TB among Malaysian aborigines..." can't be answered by a case-control design. Again case-control study would start from smear conversion staus and look out for factors. In your study diagram (figure 1) all aborigines diagnosed with TB between 2016-2020 were included and there was no sampling involved. In this case you can remove your sampling section including sample size calculation and just provide the description of that (that all were included, no sampling applied etc). I, therefore, suggest removing the case-control study design from throughout the paper and just provide the description of the data and how that is captured into your study. 

Response 1:

We acknowledge the comment and suggestion by the reviewer. Due to the importance of findings to explain the epidemiology of TB among Malaysian aborigines, we decided to maintain the study objective by further elaboration and improve some clarification as below:

1) Subject recruitment (section 2.1), on Line 84-90

This study was conducted from December 2020 until the end of May 2021 utilizing secondary data from the MyTB surveillance system from the TB and Leprosy Control Sector, Ministry of Health Malaysia. The study involved two parts. In the first part, we included all TB cases among Malaysian aborigines registered in MyTB from 1st January 2016 to 31st December 2020 to explore the TB burden in Malaysian aborigines. For the second part, we conducted a case-control study to determine the factors associated with delayed sputum-smear conversion among Malaysian aborigines. We included …

2) Sample size calculation (section 2.2). (Line 104).

…. for the case-control study …

3) Following that, we redefine the caption for Figure 1 as ‘The flowchart for the case-control study” (Line 100 and Line 201)

Point 2: I think this section (lines 253-255) needs rewriting - "The majority of patients with diabetes mellitus (59.6%), HIV co-infection (57.1%), advanced pre-treatment chest x-ray findings (66.7%), and not adhering to treatment (93.2%) had delayed sputum- smear conversion". It is incorrect because this study didn't study diabetic patients, HIV patients but it rather quantified diabetes and HIV among TB patients. So better to present the comparison of the proportion with diabetes or HIV among delayed converters and the controls. 

Response 2:

The statement was improved as below: (Line 234-238).

The majority of cases with delayed sputum-smear conversion had diabetes mellitus (59.6%), HIV coinfection (57.1%), advanced pre-treatment chest x-ray findings (66.7%), and not adhering to treatment (93.2%) compared to the controls group.

Point 3: Is the HIV status in the logistic regression model adjusted for ART? if not please include in limitation of the study

Response 3:

The ART status was not included in the study. We agree with the recommendation given. Therefore, we add it in section 4.4 (Study Limitations) as below: (Line 406-408).

The statement: “Besides that, the antiretroviral therapy (ART) status among HIV patients were not included in the study, thus, the logistic regression model analysis were not adjusted for ART”

Point 4: Interpretation of results needs to based on the analysis. For example- On lines 471-473 the authors interpreted the odds ratio as - Malaysian aborigines with smear-positive pulmonary TB who smoked were 3.25 times more likely to have delayed sputum-smear conversion than non-smokers when other confounders were controlled. That is not quite right. This should be interpreted something like - the odds of smoking among individuals with delayed smear conversion is about 3.2 times that among controls. Please correct others to be consistent with your analysis (Example line 507)

Response 4:

The statement was improvised to give the right interpretation of results by comparing risk factors between case and control group as below:

“The odds of smoking among individuals with delayed sputum-smear conversion was about 3.25 times that among the controls group when other confounders were adjusted” (Line 327 – 329).

“The current findings showed that Malaysian aborigines who had underlying diabetes mellitus with delayed sputum-smear conversion had the odd of 12.84 times than those controls group when other confounders were adjusted” (Line 346 – 348).

“Current research revealed that the odds of HIV co-infection among individuals with delayed sputum-smear conversion is about 9.76 times that among the controls group when other confounders were adjusted.” (Line 363-365). 

Reviewer 3 Report

Thanks to the amendments, the article improved the quality of the results and their presentation.

Author Response

We appreciate all the comments and recommendations given. Thank you.